# A Machine Learning Model to Predict Cardiovascular Events during Exercise Evaluation in Patients with Coronary Heart Disease

**DOI:** 10.3390/jcm11206061

**Published:** 2022-10-14

**Authors:** Tao Shen, Dan Liu, Zi Lin, Chuan Ren, Wei Zhao, Wei Gao

**Affiliations:** 1Department of Cardiology, Peking University Third Hospital, National Health Commission Key Laboratory of Cardiovascular Molecular Biology and Regulatory Peptides, Beijing 100191, China; 2Aerospace Information Research Institute, Chinese Academy of Sciences, Beijing 100094, China; 3Physical Examination Center, Peking University Third Hospital, Beijing 100191, China

**Keywords:** coronary heart disease, cardiopulmonary exercise testing, exercise safety, exercise risk precaution, machine learning

## Abstract

Objective: To develop and optimize a machine learning prediction model for cardiovascular events during exercise evaluation in patients with coronary heart disease (CHD). Methods: 16,645 cases of cardiopulmonary exercise testing (CPET) conducted in patients with CHD from January 2016 to September 2019 were retrospectively included. Clinical data before testing and data during exercise were collected and analyzed. Results: Cardiovascular events occurred during 505 CPETs (3.0%). No death was reported. Predictive accuracy of the model was evaluated by area under the curve (AUC). AUCs for the SVM, logistic regression, GBDT and XGBoost were 0.686, 0.778, 0.784, and 0.794 respectively. Conclusions: Machine learning methods (especially XGBoost) can effectively predict cardiovascular events during exercise evaluation in CHD patients. Cardiovascular events were associated with age, male, diabetes and duration of diabetes, myocardial infarction history, smoking history, hyperlipidemia history, hypertension history, oxygen uptake, and ventilation efficiency indicators.

## 1. Introduction

Exercise-based cardiac rehabilitation can reduce the mortality rate of patients with coronary heart disease (CHD), promote health, and improve clinical outcomes [1]. However, improper or extensive exercise may lead to an elevated cardiovascular risk. Cardiopulmonary exercise testing (CPET) uses gas monitoring technology and bicycle ergometer or treadmill exercise technology. CPET can detect the real-time changes of oxygen uptake, ventilation efficiency, heart rate, and other parameters [2]. Through the analysis of the above parameters in the process of exercise, it is possible to provide precaution for exercise evaluation events and to avoid risks in time.

Cardiac rehabilitation under the guidance from exercise evaluation is safer [3]. Exercise evaluation is also relatively safe in CHD patients [4], but there are insufficient evidences on risk precaution [5]. Developing prediction models for predicting cardiovascular events during exercise evaluation in CHD patients based on multi-dimensional clinical information is in urgent need clinically, which can provide early precaution of cardiovascular events and improve the safety of exercise evaluation. However, it has not yet been reported in previous literature. Due to the large number of clinical predictors and their interactions, traditional regression analysis may be unstable. In recent years, machine learning technology has been recognized as a robust tool for risk evaluation. This study aims to develop multiple machine learning-based prediction models and compare their performance so as to explore the optima reliable l algorithm for predicting the risk of cardiovascular events during exercise evaluation in CHD patients.

## 2. Materials and Methods

### 2.1. General Characteristics

The study was approved by the Peking University Third Hospital Ethics Committee (approval number 2020113). Due to its retrospective nature of observation, the ethics committee waived the need for informed consent. ClinicalTrials.gov ID was NCT04963478.

A total of 16,645 CHD patients who underwent CPET from January 2016 to September 2019 at heart canter of Peking University Third Hospital were retrospectively included in the study. CHD was defined as history of angiographically sever coronary artery stenosis (>70% diameter stenosis of any epicardial coronary artery), angina pectoris, myocardial infarction, or coronary revascularization. Clinical data of patients collected before the testing included general information (such as gender, age and disease diagnosis), past history, medication history and family history (such as history of hypertension, diabetes, hyperlipidemia and other risk factors for CHD, history of chronic obstructive pulmonary disease and other respiratory diseases, duration and medication history of the above diseases, and family history of CHD), personal history (such as smoking history and exercise habits) and pre-test examination (such as resting blood pressure, resting heart rate, and body mass index).

### 2.2. Cardiopulmonary Exercise Testing

Ultima CardiO2 CPET system was used for exercise testing. The multistage protocol with bicycle ergometer was adopted. Bruce protocol was adopted in the treadmill. Blood pressure, ECG, and symptoms of patients were collected during exercise testing. Patients were required to exercise with symptomatic limitations (respiratory exchange rate ≥ 1.1). All patients were observed under no workload for 5 min after the exercise. The whole procedure was performed under ECG monitoring, accompanied by a professional physician.

During the exercise testing, symptoms, blood pressure, heart rate, oxygen uptake, and other information of CHD patients were continuously recorded. Indicators directly collected included oxygen uptake, heart rate, oxygen pulse, power and ratio of the above indicators to the predicted value. They were further calculated to get the following indicators: oxygen uptake efficiency slope (OUES) and ventilation per carbon dioxide output slope (VE/VCO_2_ slope) [2].

Data of exercise-related cardiovascular events were recorded. Exercise-related cardiovascular events can be classified into the conditions as follows [6]: (1) angina pectoris; (2) frequent premature ventricular contractions (PVCs); (3) atrial arrhythmia, including atrial flutter, atrial fibrillation, atrial tachycardia, frequent premature atrial contractions and paroxysmal supraventricular tachycardia; (4) ventricular tachycardia (NSVT); (5) bradyarrhythmia, such as bundle branch block, sinus arrest and atrioventricular block; (6) more than 10 mmHg drop in blood pressure; (7) other events leading to the CPET termination (Figure 1). Positive ECG criterion was ST segment down or horizontal sloping depression for 80 ms of ≥0.10 mV. ST segment persistently elevation >0.10 mV after the J point was also considered an abnormal response [7].

### 2.3. Statistical Methods

A panel of cardiac cardiologists and epidemiologists was set up for the study. Before the study began, the panel developed standard searching strategies to avoid inconsistency. CHD patients were divided into two groups, patients with CPET-related cardiovascular events in one and the rest in the other. Whether exercise-related cardiovascular events occurred was used as the dependent variable, and various clinical information of patients as independent variables, either continuous or discrete. In the data set, a few data (less than 5%) were incomplete. Based on complete data of CHD patients, we replaced incomplete data by the medians of corresponding data. In order to explore the correlation between independent variables and dependent variables, we used machine learning methods to develop, train and analyze prediction models so as to obtain rankings of correlations between independent variable and various dependent variables. Finally, the risk of exercise-related cardiovascular events in CHD patients was predicted using the above models.

Python 2.7 (Anaconda) platform and scikit-learn 0.19.1 framework were used to train the model, making it possible to develop machine learning frameworks in experiments. With the above platforms, four machine learning model types targeting problems in this study were developed and compared: support vector machine (SVM), gradient boosting decision tree (GBDT), extreme gradient boosting (XGBoost), and logistic regression (LR). These four algorithms had good robustness and generalization ability, and were recognized as classical machine learning algorithms for performing classification tasks.

We used a method called 10-fold cross-validation to compare the performance of algorithms above. First, the data (the 73 selected features) were randomly divided into ten folds, which was applied in ten experiments. In every trial, we used one set of data as the test set and the remaining data as the training set. And then, we trained the model and calculate the accuracy. At the end of the ten experiments, we calculated the average of the accuracy as the accuracy of each model. By the validation above, we obtained the area under the curve (AUC) of each model.

LR mean a generalized logistic regression model in our study. This method was very common and effective in the field of disease diagnosis and data mining [8]. First, binary logistic regression was performed on all variables. Multivariate analysis was then conducted to further explore their association with the occurrence of events. Predictive accuracy and ROC curve were used to evaluate the effectiveness of the final model.

SVM was a technique based on statistical learning method [8]. This method tried to find an optimal classification hyperplane in the feature space, on the premise of ensuring accuracy and classification requirements at the same time. Before the emergence of deep learning, SVM was one of the most popular classification models. Theoretically, it can achieve optimal classification of linearly separable data.

GBDT was the most suitable method for fitting the real distribution in machine learning. It was an ensemble algorithm based on decision tree, which iterates the learners by gradient descent. GBDT can achieve better classification results than a single learning period by combining multiple learners. GBDT had the advantage of good generalization performance. In addition, it can handle various types of data flexibly and achieve high prediction accuracy.

XGBoost was a modified form of GBDT technique [9]. GBDT algorithm uses classification and regression tree. XGBoost can not only supported linear classifiers, but also handled CART classifiers, which make XGBoost get simpler model and prevent overfitting of the model. Moreover, XGBoost used both the first and second derivatives for optimization, performs second-order Taylor expansion to minimize the loss function, whereas GBDT only used first derivative information. In addition, XGBoost allowed custom optimization objectives and evaluation criteria, as long as the function was differentiable in the first and second order. More importantly, XGBoost had built-in rules for handling missing values. For a sample with a missing feature value, XGBoost can automatically learn the splitting direction.

Scikit-learn 0.19.1 was used to conduct LR, SVM, and GBDT algorithms. XGBoost algorithm was implemented using XGBoost 0.82 framework combined with scikit-learn 0.19.1. In terms of setting, LR adopted L1 regularization. SVM adopted linear kernel. The remaining parameters of all algorithms were set by default.

## 3. Results

### 3.1. General Information of Patients

A total of 16,645 cases of CPET performed by 16,645 unique individuals were included. The mean age of CHD patients was 57.5 ± 12.9 years old, and there were 11,647 males (69.9%). 52.7% of patients were complicated with hypertension, 24.0% with diabetes, and 51.8% with hyperlipidemia (Table 1).

The majority of patients (*n* = 14,826, 89.1%) completed symptom-limited CPET with termination resulting from symptoms of fatigue or dyspnea. The remaining 10.9% of patients discontinued testing because of ECG changes or exercise-related cardiovascular events. Table 2 described the stress test results. Effort was generally good. The mean peak RER 1.12 ± 0.12. About 1782 patients (10.7%) had an exercise ECG positive, meaning for ischemia.

### 3.2. Development of Prediction Models for Cardiovascular Events during CPET in CHD Patients

CPET-related cardiovascular events occurred in 505 (3.0%) enrolled patients, including 63 of angina attack, 247 of frequent PVCs, 58 of NSVT, 104 of atrial arrhythmia, 21 of bradyarrhythmia, and 5 of sharp drop in blood pressure during exercise. No death was reported.

In this study, the dataset was randomly split into feature selection set (80%) and test set (20%). The feature selection set was used for model training and feature ranking. The feature selection set was further randomly split into training and validation sets, which accounted for 50% and 30% of the dataset respectively. The test set was used to validate the performance of the prediction model.

The research flowchart was shown in Figure 2. First, clinical information of patients was preprocessed. The training set was then used for model training. The correlation ranking of all 73 features was obtained (Table 3 and Appendix A) based on the cumulative results of ten experiments. In the fitting process of the regression model, higher dimension of the feature set will increase the complexity of the model. In addition, some of the original features may demonstrate redundance and collinearity. Therefore, it is necessary to conduct feature selection. In this study, features with worst performance in comprehensive medical analysis and correlation ranking were removed successively from the experiment.

In addition, we found prediction models developed by different machine learning methods varied in performance. Among them, XGBoost showed the highest accuracy (Figure 3), owing to its advantage in preventing overfitting and taking into account sparse data. Therefore, this study used the XGBoost algorithm for subsequent analysis of the selected features.

By observing the ROC curves, the XGBoost has the best AUC index performance, owing to its advantage in preventing overfitting and taking into account sparse data.

We used the feature selection set to train the model and test set to test the model. Based on the importance ranking of features obtained by machine learning methods and results of medical analysis, features with poor prediction effects were successively removed, and the changes of accuracy (AUC) of prediction models with different feature sets were recorded. Based on the cumulative results of ten experiments, curves of AUC’s response to feature selection were shown (Appendix A). Through analysis, we found that the XGBoost machine learning model still showed good stability and accuracy after removing 50 features. It suggested that original features had redundancy, and the reducing dimension of features did not degrade the accuracy of the model.

With further analysis of the accuracy results, an effective and low-dimensional prediction model was finally obtained. As shown in Figure 4, with continuous removal of features, the model based on only ten features still showed reliable performance (AUC = 0.8115), and there was no significant difference from prediction results of the model with high-dimensional features. We obtained a high-quality, lightweight prediction model. Features included are age, male, diabetes and duration of diabetes, myocardial infarction history, smoking history, hyperlipidemia history, hypertension history, VE/VCO_2_ slope and VO_2_@AT.

The machine learning model based on ten features demonstrated relatively reliable performance (AUC = 0.8115).

## 4. Discussion

Exercise risk evaluation has important clinical benefits for CHD patients. Symptom-restricted CPET can provide an objective and quantitative evaluation of the overall performance status for those patients. Exercise evaluation is relatively safe for CHD patients, but the research on risk prediction is insufficient. Timely identification and early precaution of exercise-related cardiovascular events during exercise is an important means to ensure exercise safety for CHD patients. Targeting CHD patients, this study included important clinical indicators, and developed a risk prediction model for cardiovascular events based on CPET indicators.

### 4.1. Advantages of Machine Learning Methodology Based on Feature Selection

Patient management and disease prediction can be defined as “pattern recognition” tasks, which has been a consensus in the academic community. Therefore, the development of model fitting and prediction methods based on machine learning has also facilitated progress in the field of disease prediction [10]. Compared with traditional statistical methods, machine learning models are better at handling high-order interactions between features and prediction outcomes, and are more applicable for cases with a large number of features in medical field [11,12]. Additionally, machine learning technology is more flexible. Given the correct parameters and performance standards, machine learning can develop a better solution by optimization and combining multiple basic methods.

In the study, the performance of several machine learning methods in predicting cardiovascular events in patients during exercise evaluation was tested. The original clinical data were collected before and during exercise testing, which were further used for model training and validation. Results proved machine learning to be a very suitable prediction tool for cardiovascular events in patients during exercise evaluation. Most algorithms can achieve an accuracy of 0.7 or higher. Especially, XGBoost showed the best performance. Compared with other algorithms, XGBoost can reduce overfitting by incorporating regularization and tree pruning, and achieve stronger classification capability by combining weak learners [13]. In addition, XGBoost takes into account data sparsity caused by missing data, which brings about better performance [14].

Feature engineering has a crucial impact on the performance of machine learning models. Feature selection determines the upper limit of prediction model performance, and different prediction methods are constantly approaching this upper limit. High-dimensional features, such as 73 features in this study, will increase the complexity of the prediction model and result in poor robustness and generalization ability of the model, thereby degrading its performance. In order to achieve high-quality feature selection, this study analyzed 73 original features one by one, gradually reduced the feature dimensions and the complexity of the prediction model by removing features with poor performance, and evaluated the accuracy of models. After feature ranking and experiments, this study developed a model of ten features with reliable prediction results (AUC = 0.8115). Based on the existing CPET database, the prediction model developed in this study can predict risks of exercise-related cardiovascular events in real time, which is helpful for real-time precaution of cardiovascular events and avoiding risks by reasonable interventions.

### 4.2. Important Predictive Value of Multi-Dimensional Clinical Information Based on CPET

Detailed clinical data collection before exercise testing is helpful for exercise risk evaluation. Through analysis of patients’ baseline data before exercise testing, this study found that older patients, male, and patients with diabetes, hyperlipidemia, hypertension, smoking history were at higher risk of exercise-related cardiovascular events. Previous studies have also suggested significant age and sex differences in the incidence of exercise-related cardiovascular event [15]. Older patients are at higher cardiovascular risk during exercise evaluation due to decreased cardiac reserve and ventilation efficiency. Sex difference may result from higher exercise intensity of male than female during exercise or exercise testing, which may be associated with higher cardiovascular risk.

Results suggest that having coronary heart disease with diabetes and longer duration of diabetes may increase cardiovascular risks during exercise. The possible mechanism may be that long-term hyperglycemia leads to energy metabolism disorder and then causes heart systolic and diastolic dysfunction and chronotropic incompetence [16]. European Society of Cardiology recommends that diabetes patients undergo cardiovascular safety evaluation before exercise. The valuation included glycemic control, comorbidities, and treatment options [17].

It is necessary to pay attention to symptoms of patients during exercise, such as new-onset chest pain, palpitations, dizziness and shortness of breath, and further conduct real-time analysis of gas metabolism indicators during exercise, which may provide immediate precaution of cardiovascular events during exercise evaluation. These indicators include oxygen uptake-related indicators which reflect left ventricular output, and ventilation efficiency indicators which reflect right ventricular function [18,19]. Among them, VE/VCO_2_ slope is an important index of ventilation efficiency. It reflects the ratio of minute ventilation to carbon dioxide production. The normal value was 20–30. When the ratio increases, there is an inconsistent ventilation-perfusion relationship, and the curve slope increases abnormally [20]. Real-time calculation and display of ventilation efficiency during exercise will be helpful for identification of cardiac risks during exercise testing [21].

### 4.3. Application of the Prediction Model

This study constructed a machine learning-based prediction model for cardiovascular events in exercise testing based on CPET indicators. Features included are age, male, diabetes and duration of diabetes, myocardial infarction history, smoking history, hyperlipidemia history, hypertension history, oxygen uptake parameter and ventilation efficiency. The prediction model has good application prospects. First, reasonable use of sensitive indicators to provide dynamic precaution of cardiovascular events during exercise evaluation in people undergoing exercise rehabilitation. Second, exercise risk score of patients can be used to assess the cardiovascular risk during exercise-based rehabilitation, which can provide an important basis for developing safe and effective exercise schemes.

### 4.4. Research Limitations

However, there are several limitations. It is a retrospective single center study. Further validation and generalization of results in more centers are needed. There are limitations with the database to describe the study population. A few patients did not undergo testing, such as electrocardiogram and echocardiography in close proximity to CPET. In addition, we did not collect information on future cardiovascular events in CHD patients. Prospective studies are needed to explore methods for precaution during exercise rehabilitation with telehealth equipment.

## 5. Conclusions

Machine learning methods (especially XGBoost) can effectively predict cardiovascular events during exercise evaluation in CHD patients. CPET-related cardiovascular events were associated with age, male, diabetes and duration of diabetes, myocardial infarction history, smoking history, hyperlipidemia history, hypertension history, oxygen uptake parameter, and ventilation efficiency. Our results support the identification of cardiovascular events during exercise testing and exercise training in CHD patients.

## Figures and Tables

**Figure 1 jcm-11-06061-f001:**
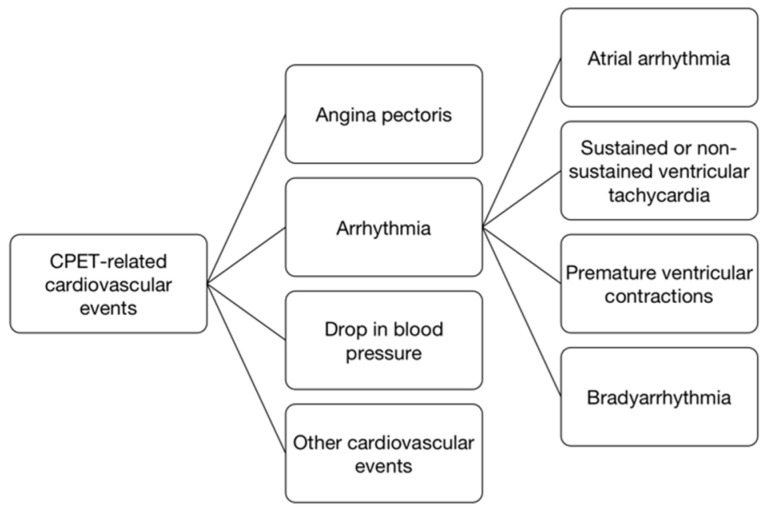
Tree diagram summarizing CPET-related cardiovascular events. CPET, cardiopulmonary exercise testing.

**Figure 2 jcm-11-06061-f002:**
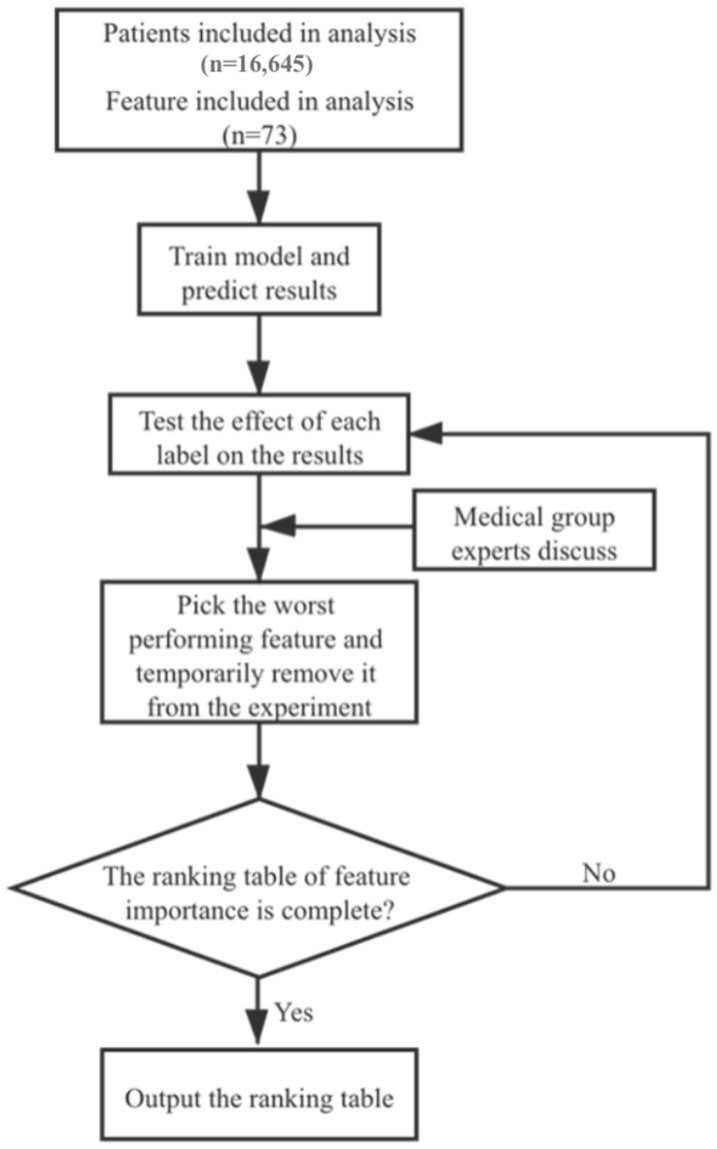
Flowchart of feature selection.

**Figure 3 jcm-11-06061-f003:**
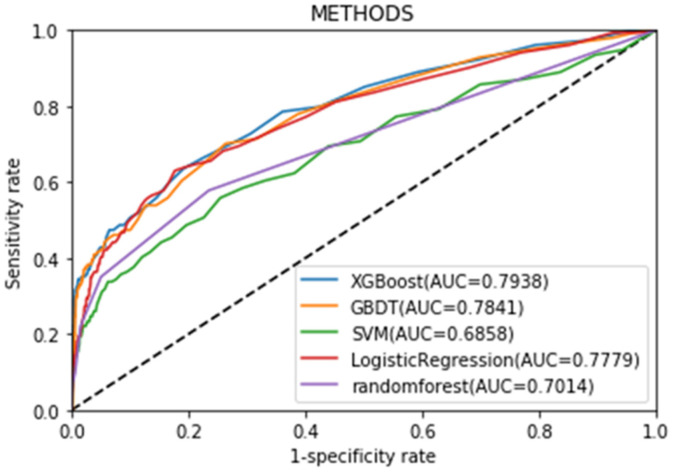
ROC curve of different machine learning prediction models. AUC, area under curve.

**Figure 4 jcm-11-06061-f004:**
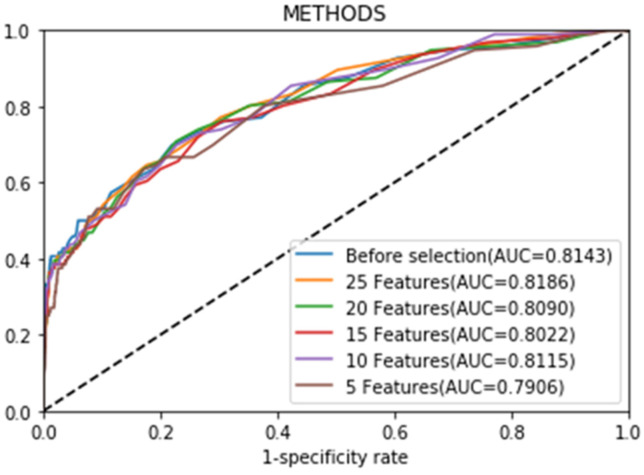
AUC curve of prediction models with different numbers of features. AUC, area under curve.

**Table 1 jcm-11-06061-t001:** General information of CHD patients.

Parameters(%), Mean ± SD	Value
Age (years)	57.5 ± 12.9
Male, N (%)	11,647 (69.9)
Baseline BMI (kg/m^2^)	25.5 ± 3.4
Hypertension, N (%)	8786 (52.7)
Hyperlipidemia, N (%)	8637 (51.8)
Diabetes, N (%)	3999 (24.0)
Smoking history, N (%)	7071 (42.1)
Family history of CHD, N (%)	4562 (27.4)
Exercise habit, N (%)	10,454 (62.8)

Abbreviation: BMI, body mass index. CHD, coronary heart disease.

**Table 2 jcm-11-06061-t002:** CPET results.

Parameters	Value
VO_2_peak (mL·kg^−1^·min^−1^)	21.4 ± 6.3
VO_2_peak/Pred (%)	69.8 ± 10.4
RERpeak	1.12 ± 0.12
HRpeak (bpm)	136 ± 23
O_2_-pulse peak (mL/beat)	11.4 ± 3.2
SBPpeak (mmHg)	171 ± 21
VO_2_@AT (mL·kg^−1^·min^−1^)	15.9 ± 5.8
HR@AT (bpm)	109 ± 15
O_2_-pulse@AT (mL/beat)	8.4 ± 2.6
Resting HR	73.9 ± 15.4
Resting SBP (mmHg)	127 ± 19
VE/VCO_2_slope	28.7 ± 4.2

Abbreviation: CPET, cardiopulmonary exercise testing. VO_2_peak, peak oxygen uptake; VO_2_peak/Pred, ratio of peak oxygen uptake to predicted; RERpeak, peak respiratory exchange ratio; HRpeak, peak heart rate; O_2_-pulse peak: peak oxygen pulse; SBPpeak, peak systolic blood pressure; VO_2_@AT, oxygen uptake at anaerobic threshold; HR@AT, heart rate at anaerobic threshold; O_2_-pulse@AT, oxygen pulse at anaerobic threshold; VE/VCO_2_slope, ventilation per carbon dioxide output slope.

**Table 3 jcm-11-06061-t003:** Importance ranking of top 10 clinical features included in machine learning.

	Feature Name	Total Score
1	Age	730
2	Duration of diabetes	662
3	Diabetes history	662
4	Myocardial infarction history	658
5	Male	642
6	VE/VCO_2_slope	630
7	Smoking history	616
9	Hyperlipidemia history	552
8	VO_2_@AT	564
10	Hypertension history	528

Abbreviation: VE/VCO_2_slope, ventilation per carbon dioxide output slope; VO_2_@AT, oxygen uptake at anaerobic threshold.

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
