# Peer review of "A Machine Learning Model to Predict Cardiovascular Events during Exercise Evaluation in Patients with Coronary Heart Disease"

_jcm, 2022, doi:10.3390/jcm11206061_

Round 1
Reviewer 1 Report
The purpose of the manuscript is to use machine learning algorithms to predict cardiovascular events during exercise assessment in patients with CHD. Authors claimed to have better performance using XGBoost. They have used real patient data but the experiment has several limitations too. Authors need to do a better job at literature survey, experimental design explanation results in comparison, etc. In my opinion, the manuscript is suitable for publication in the “Journal of Clinical Medicine” only after solving the mentioned major issues.
Major points:
1. Authors mentioned in line 43 about insufficient studies on risk prediction using CPET, there are several research done using CPET and risk prediction, and authors need to mention those along with their performances.
2. Provide a tree diagram summarizing CPET-related cardiovascular events classification (line 93).
3. What is the rationale behind choosing the four algorithms? (line 126) (why other ML algorithms were not chosen to compare)
4. How did the authors make sure that those were independent variables and not strongly correlated with each other? (line 116)
5. (line 198) Correlation gives linear dependency, what about nonlinear dependency among features?
6. How did the author make sure about removing redundant features? The author needs to better describe their approach.
7. Figure 1, how did the author figure out the worst performing feature? What is the index for that? How did the author know the ranking table of feature importance is complete or not? The author needs to clarify those and make figure 1 more understandable.
8. To clarify figure 1, the authors are suggested to provide an example with a feature with values to follow through the whole flow chart.
9. “with further analysis of accuracy results .. .. obtained” (line 227), what is the “analysis” here? how a low dimensional prediction model was obtained?
10. From 73 to 10 features, the author needs to clarify how these 10 features may or can replace the use of those 73 in terms of biological significance too. Just removing it due to math should not be enough rationale.
11. Authors need to elaborate on section 4.4 (limitations), the manuscript and experiment contain more limitations than they have listed here, also along with limitations the authors are suggested to provide probable solutions for those.
12. Also, the authors need to provide a table comparing the performance of their experiment with other contemporary experiments and results from recently published articles.
13. Cardiovascular disease risk prediction using ML is a very common research topic, although authors have used CPET authors need to spend a paragraph describing how their research is different from others and how it will provide additional value to the topic. This is very important.
Reviewer 2 Report
The paper by Shen et.al described that the supervised machine learning methods, including LR, SVM, GBDT, and XGBoost, were useful to predict CV events occurring in the CPET performed in 16645 patients with known coronary artery disease (CAD). After assessing feature importance for all variables (clinical data and CPET-related parameters, in total 73 features) to make 10 features with the highest discriminative power (AUC = 0.81), the authors incorporated them into the four ML algorisms using ten-fold cross-validation to produce the mean of accuracy. Of these, XGBoost, an improved version of Gradient Boosting framework using parallel tree boosting instead of sequential one, have shown the best predictability for CPET-related CV event.
The analytical method was interesting and the result (XGBoost defeated the others) was reasonable; however, the following issue should be addressed.
1) Although the data analysis automatically demonstrated both the presence of diabetes and duration of diabetes were selected to be significant features for the prediction model, it is questionable for potential collinearity. Rather, more predictive baseline variables, such as ejection fraction or abnormal Q waves were missing in the original features. These could be readily obtained before CPET.
2) Process of ML analysis using python and scikit-learn framework cannot be evaluated as to whether it is scientifically fair or not. It appears fair, but not transparent for review. Please consider to demonstrate with more transparency. Attaching cords via GitHub may be one idea (this can be submitted only for review process if it is unfavorable).
Reviewer 3 Report
The manuscript entitled “A machine learning model to predict cardiovascular events during exercise assessment in patients with known or suspected coronary heart disease” was reviewed. This study attempted to develop a prediction model for cardiovascular events during exercise assessment in patients with coronary heart disease (CHD) using machine learning methods. This manuscript is very interesting and well written. However, there are some issues that should be properly addressed.
1. The reviewer suggests the predictors of cardiovascular events during exercise testing should be shown as a Table, not a supplement Table (at least the top 10 predictors).
2. Although the authors tried to investigate predictors of CPET-related cardiovascular events, the results of CPET (such as VE/VCO2slope, VO2@AT, etc.) were included in the outcomes. The reviewer wonders clinicians cannot obtain the information before CPET. If the authors aimed to apply these outcomes to the predictors of cardiovascular events during exercise rehabilitation, it should be described clearly.
Round 2
Reviewer 1 Report
Authors have made extensive revisions over their initial draft manuscript. The following issues need to be taken care of before its suitable to be published.
1. Reply for point number 3 (the rationale behind choosing 4 algorithms) is not satisfactory. Authors need to either provide references as evidence of using these four algorithms as standard or prove why they are better than other algorithms. Stating that they are mainstream is not sufficient.
2. In point number 4 authors corrected their mistake about using “independent” characteristics for all variables in the dataset. The question remains, have authors calculated how strongly correlated those with each other?
3. Answer to number 5 is not satisfactory. Correlation measures linear dependency. The authors need to explain did they observe any nonlinear dependency among variables. Yes, or no? if yes then how did they intend to solve that issue? If not, then what is their proof behind this conclusion?
4. Point number 6, what is the threshold for removing the variables? Just stating ‘variables were removed’ is not enough.
5. Point number 7, importance ranking implementation by machine learning is not quite a clear statement. Specify which machine learning algorithm, and how/why that machine learning algorithm is best at ranking features.
6. Point number 10, “accuracy loss of whole prediction model was not so high”, authors need to provide values not just statements. What was the loss threshold for removing features from 73 to 10?
